# E-CommerceVideo: A Benchmark and Approach for E-Commerce Video Generation from Product Images

## Abstract

We introduce the task of *E-Commerce Video Generation*, which aims to automatically produce dynamic product showcase videos from static product images and a text prompt describing the video's motion and background. Unlike general video generation, this task requires strict adherence to the product's identity and visual features. Even small distortions in color, texture, or logos are not acceptable for commercial use. Existing methods are not directly applicable, as single-image conditioning often leads to hallucinations of product details, while strict frame-based conditioning severely limits motion diversity. To address this gap, we present **E-CommerceVideo**, a large-scale, multi-modal dataset curated from the Taobao product repository. This benchmark dataset comprises a massive collection of triplets: product images, corresponding textual descriptions, and high-quality videos. We further establish a simple yet efficient **baseline method** by adapting a pre-trained video generation model with a VAE-based spatial injection mechanism, which preserves product appearance while enabling motion synthesis. Comprehensive evaluation on VBench demonstrates the baseline's potential as an entry point for subsequent advancements.

## 1 Introduction

The rapid growth of e-commerce has intensified the demand for engaging product showcases. While static product images remain standard, videos have proven significantly more effective in attracting attention, demonstrating features, and increasing conversions. However, producing professional videos requires costly shooting and editing, creating a major bottleneck for merchants with large inventories. Automated video generation offers an appealing alternative, yet existing solutions fall short in this domain.

Despite the progress in general video generation, existing methods face significant limitations in the e-commerce domain, primarily due to the core challenge of maintaining strict fidelity to the product's intricate details and identity while generating attractive, dynamic showcase motions. Standard Image-to-Video (I2V) approaches Wan et al. (2025); Sor (2024); kel (2024), for instance, use the first frame as a strict conditioning signal, which severely limits the flexibility and freedom of the generated motion and background. On the other hand, methods relying on a single reference image Zhang et al. (2023); Hu (2024) lack comprehensive visual information from multiple perspectives, often causing the model to hallucinate or alter crucial product details (e.g., changing clothing patterns) when generating motion, leading to unacceptable inconsistencies for commercial use.

To address these challenges, we first construct the E-CommerceVideo dataset, a large-scale, multi-modal benchmark curated from the Taobao product repository. It comprises a massive collection of triplets—multi-view product images, textual descriptions, and high-quality video demonstrations—providing the necessary data foundation for this task. Alongside the dataset, we introduce a **baseline method** built on a pre-trained video generation model with VAE-based spatial encodings. Although simple by design, this baseline highlights the challenges of preserving product fidelity while generating dynamic showcases and provides a starting point for future improvements. Our main contributions are as follows:

Figure 1: Examples of generated e-commerce videos with reference images and text.

- We introduce **E-CommerceVideo**, a large-scale, multi-modal dataset curated from the Taobao repository, offering the first benchmark dedicated to e-commerce video generation.
- We propose a baseline method that integrates product-specific information into a pre-trained video generator, serving as a starting point for future research.
- Extensive experiments are conducted using the VBench benchmark. Both quantitative and qualitative results demonstrate the effectiveness of our proposed approach, showcasing its ability to generate coherent, high-fidelity, and appealing e-commerce videos compared to existing off-the-shelf generation models.

## 2 RELATED WORK

**Image-to-Video generation**. I2V aims to automatically create temporally coherent and semantically plausible videos from one or more images. Current I2V approaches can be broadly categorized into two classes. The first class is **Keyframe-to-Video** Guo et al. (2023); Chen et al. (2023); Wang et al. (2023b), where the input image serves as the initial frame of the video, and the model generates subsequent frames that naturally extend from it. For example, Stable Video Diffusion Blattmann et al. (2023) treats the input image as frame 0 and predicts subsequent frames sequentially using temporal attention or 3D convolutions. However, due to the lack of global modeling, these methods often suffer from identity drift or structural distortion when generating long videos. To mitigate this issue, Sora Sor (2024) and other DiT-based architectures Wan et al. (2025); kel (2024); Zheng et al. (2024) expanded the image encoding into a spatio-temporal latent sequence and generated the entire video in one pass using Transformers, thereby effectively preserving cross-frame identity and content consistency. The second class is **Reference-Image-to-Video**, where the input image serves as a prior for identity, structure, or style, guiding the generation of dynamic videos that maintain semantic consistency while allowing free starting points and camera motions. For instance, AnimateAnyone Hu (2024) extracts identity features such as facial information from the reference images and imposes feature alignment constraints on every generated frame to ensure consistent appearance, clothing style, and motion changes. Meanwhile, I2VGen-XL Zhang et al. (2023) generates videos from a reference image using a two-stage diffusion framework: first, a dual-encoder architecture fuses global semantics and local details to produce a low-resolution video; then, a text-conditioned super-resolution stage refines it into a high-definition, temporally coherent video faithful to the input image. Despite significant progress in generation quality and controllability, reference-guided I2V still faces challenges such as a gradual loss of key visual attributes from the reference image over time, and motion trajectories lacking logical or physical plausibility.

**Video Generation Dataset**. The development of video generation datasets has evolved from small-scale, manually annotated corpora to large-scale, weakly supervised data collected from the Web. Early efforts such as MSVD Chen & Dolan (2011) and MSR-VTT Xu et al. (2016) cover general open-domain content and are primarily designed for fundamental video captioning tasks. LSMDC Rohrbach et al. (2015) focuses specifically on the movie domain, advancing modeling of narrative content, while DiDeMo Anne Hendricks et al. (2017), built upon Flickr user videos, explores alignment between unstructured short clips and textual descriptions. With the rapid advancement of generative models, domain specificity has emerged as a new trend. HT100M Miech et al. (2019) targets instructional or tutorial videos, constructing a corpus of hundreds of millions of tutorial clips. ActivityNet Caba Heilbron et al. (2015) Captions concentrate on action-centric scenarios, enabling complex temporal reasoning. WebVid-2M Bain et al. (2021) and InternVid Wang et al. (2023c) re-

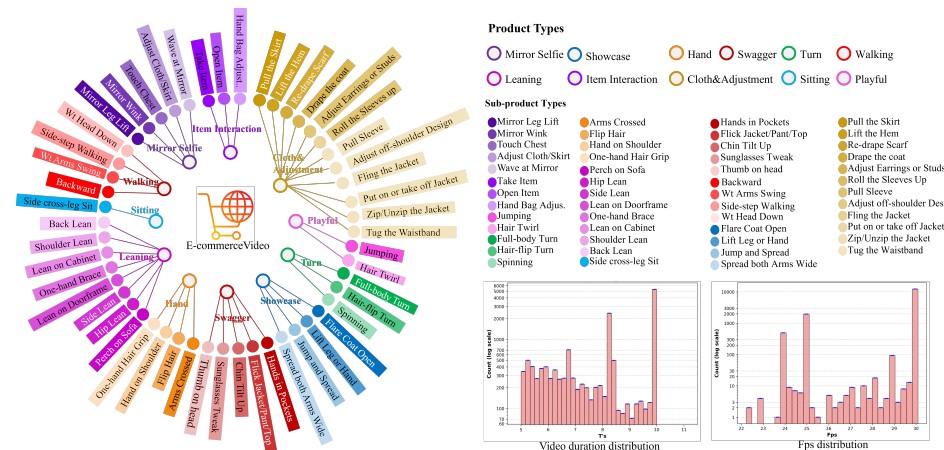

Figure 2: The two-level semantic structure of E-CommerceVideo. We collect data of 12 product types and 51 sub-product types from the Taobao product repository. We also provide statistics on video duration and frame rate distribution.

turn to the open-domain setting but employ CLIP-based filtering to ensure high-quality text–video alignment, thereby supporting diffusion-based video generation. However, existing datasets predominantly adhere to the T2V paradigm and lack dedicated resources for the e-commerce domain. Although some datasets, such as How2 Sanabria et al. (2018) and HT100M Miech et al. (2019), include product demonstrations or usage tutorials, they do not provide a video generation setup conditioned on product images. To address this gap, we introduce E-CommerceVideo, a novel dataset that not only offers real-world, multiview product images paired with corresponding demonstration videos, but also includes precise and detailed video descriptions. This work fills a critical void in current T2V and I2V research for automated visual content generation in e-commerce scenarios.

Table 1: **Comparison of E-CommerceVideo with previous video generation dataset.** Datasets are categorized based on domain. Our E-CommerceVideo data fills the gap in video generation datasets specifically for the e-commerce domain.

| dataset | domain | #clips | avg dur. (secs) | #avg sent. (words) | #images | task |
|---|---|---|---|---|---|---|
| ActivityNet Caba Heilbron et al. (2015) | Action | 100K | 180 | 13.5 | - | Video Caption |
| MSR-VTT Xu et al. (2016) | Open | 10K | 15 | 9.3 | - | Video Caption |
| LSMDC Rohrbach et al. (2015) | Movies | 118K | 4.8 | 7.0 | - | Video Caption |
| DideMo Anne Hendricks et al. (2017) | flickr | 27K | 28 | 8.0 | - | Video Caption |
| YouCookII Zhou et al. (2018) | Cooking | 14K | 316 | 8.8 | - | Video Caption |
| VATEX Wang et al. (2019) | Open | 41K | 10 | 15.2 | - | Video Caption |
| HT100M Miech et al. (2019) | Instruction | 136M | 3.6 | 4.0 | - | Video Caption |
| HD-VILA-100M Xue et al. (2022) | Open | 103M | 13.4 | 32.5 | - | T2V |
| HD-VG-130M Wang et al. (2023a) | Open | 130M | 5.1 | 9.6 | - | T2V |
| InternVid Wang et al. (2023c) | Open | 100M | 13.4 | 32.5 | - | T2V |
| OpenVid-1M Nan et al. (2024) | Open | 1M | 7.2 | - | - | T2V |
| Panda-70M Chen et al. (2024) | Open | 70.8 | 8.5 | 13.2 | - | T2V |
| MiraData Ju et al. (2024) | Open | 80K | 72.1 | 318 | - | T2V |
| **ours** | E-commerce | 15K | 8.3 | 185.6 | 75K | I2V & Video Caption |

## 3    E-COMMERCEVIDEO BENCHMARK

Previous mainstream datasets in the field of video generation primarily focused on "text-to-video" generation tasks, as shown in Table 1. These datasets spanned multiple application scenarios, such as films, cooking, and human actions, with data scales ranging from 2K to 136M, providing a solid data foundation for the rapid advancement and performance improvement of text-to-video generation models. However, in the vertical domain of e-commerce video—particularly for "reference-image-to-video" generation tasks aimed at product promotion—there remains a notable absence of systematic, high-quality, domain-specific datasets. This data gap has, to some extent, hindered the fine-grained application and commercial deployment of generative models in e-commerce scenarios.

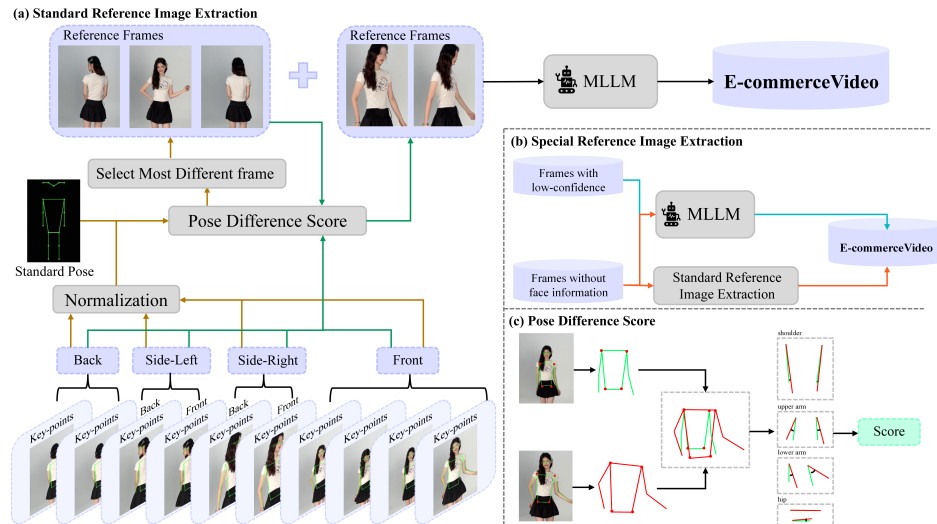

Figure 3: **Reference Images Extraction Pipeline.** (a) Reference image extraction pipeline for standard video, which showcases the model and product from multiple angles. (b) Reference image extraction pipeline for low-quality video. (c) Approach to calculating the difference between two human poses

## 3.1 E-COMMERCEVIDEO DATASET STRUCTURE

We propose the E-CommerceVideo dataset, a large-scale, high-quality dataset tailored for reference-image-to-video generation in e-commerce. E-CommerceVideo contains 15096 video-reference-images-caption triplets collected from the Taobao product repository, of which 1500 are reserved for testing and the remaining 13596 constitute the training set. Most videos have a frame rate of 30 frames per second and a resolution of 720×960. Each video sample originates from real-world e-commerce product promotion scenarios, featuring one model showcasing multiple products—primarily apparel. The videos are 5-10 seconds in duration and authentically capture the model dynamically presenting products from multiple angles within actual marketing settings, including typical promotional actions such as turning, walking, and stationary posing. As visual guidance inputs for the generation task, reference images are directly sampled and extracted from the original videos. For each video, exactly five images are uniformly selected, corresponding to key frames capturing the model from different viewpoints (e.g., front, side, back) and in diverse poses (e.g., standing display, walking motion, gesture-based presentation). This ensures that the visual guidance comprehensively and representatively covers both spatial perspectives and dynamic actions. Each video is further accompanied by a structured textual description (caption) to provide semantic-level generation guidance and content constraints. Dataset structure is shown in Figure 2.

## 3.2 E-COMMERCEVIDEO DATASET PROCESSED PIPELINE

We propose a systematic framework for constructing reference images, which is divided into two key stages: **(a) Video Frame Sampling**. **(b) Reference Images Extraction**. The framework automatically selects five visually representative and pose-diverse key frames from e-commerce videos, based on the dynamic variations of the model's poses. These frames serve as a standard reference image set for each video.

### 3.2.1 VIDEO FRAMES EXTRACTION

Since directly extracting reference images from e-commerce videos as model training input may lead the model to develop shortcut learning, we divide the e-commerce video into two parts: (a) the 0–5s video segment is used for model training; (b) the remaining video is used for reference image extraction, to ensure that reference images do not appear in the model training video. As shown in Algorithm 1, a tail-aligned strategy is adopted, sampling backward from the end of the video, ensuring that the sampling interval always avoids the first 5 seconds. By adaptively searching within

a certain frame number range and dynamically adjusting the sampling frame rate, the method ensures that videos of different lengths can all output video frame sequences with temporal continuity and semantic completeness.

After obtaining the video frame sequence, in order to ensure that the frames selected for reference image extraction contain a clear human body structure, we employ the DWpose Yang et al. (2023) model to perform human detection. For input video frame sequence:

$$\mathcal{V} = \{\mathbf{v}_i\}_{i=1}^M, \quad \mathbf{v}_i \in \mathbb{R}^{H \times W \times 3} \tag{1}$$

Dwpose Yang et al. (2023) outputs two sets of structured results: (1) Human detection bounding box set: $\mathbf{B} = \{\mathbf{b}_i\}_{i=1}^M$, where $\mathbf{b}_i = [b_{i1}, b_{i2}, \ldots, b_{iK_i}]$, and $K_i$ is the number of detected humans in frame $i$, Each bounding box $b_{ij} \in \mathbb{R}^4$. (2) Corresponding confidence score set: $\mathbf{S} = \{\mathbf{s}_i\}_{i=1}^M$, where $\mathbf{s}_i = [s_{i1}, s_{i2}, \ldots, s_{iK_i}]$, and $s_{ij} \in [0, 1]$ represents the detection confidence score for the $j$-th human in frame $i$. We define a frame quality mask $\mathbf{m} \in \{0, 1\}^M$ as follows:

$$m_i = \begin{cases} 1, & \text{if } K_i > 0 \text{ and } s_{i1} > \tau \\ 0, & \text{otherwise} \end{cases} \tag{2}$$

where $\tau$ is an empirical threshold, and $s_{i1}$ denotes the highest confidence score among all detected humans in frame $i$ (i.e., the main subject). The final filtered frame sequence is:

$$\mathcal{V}_{\text{final}} = \{\mathbf{v}_i \mid m_i = 1, \ i = 1, 2, \ldots, M\} \tag{3}$$

### 3.2.2 REFERENCE IMAGES EXTRACTION

This stage is used to select reference images from the video frame sequence $\mathcal{V}_{\text{final}}$. Each frame $\mathbf{v}_i$ needs to be fed into the pose detection model $\mathcal{M}$ to detect human keypoints. If $\mathcal{M}$ fails to detect a valid human, that is, when the confidence score of the detected human bounding box is less than a predefined threshold $\theta$, the frame will be added to a separate low-confidence set $\mathcal{V}_{\text{low-confidence}}$. If $\mathcal{M}$ successfully detect human pose, it will output 17 standard keypoints and their confidence levels:

$$\mathcal{K}_i = \left\{ (x_j^{(i)}, y_j^{(i)}, c_j^{(i)}) \right\}_{j=1}^{17}, \quad c_j^{(i)} \in [0, 1] \tag{4}$$

where $(x_j^{(i)}, y_j^{(i)})$ is the coordinate of the $j$-th keypoint, and $c_j^{(i)}$ is its detection confidence. Then, we construct a key point existence indicator vector:

$$\mathbf{e}_i = \left[ \mathbf{I}(c_j^{(i)} > T) \right]_{j=1}^{17} \in \{0, 1\}^{17} \tag{5}$$

where $\mathbf{I}(\cdot)$ is the indicator function, $T$ is the confidence threshold. Subsequently, a predefined classification function $f_{\text{cls}} : \{0, 1\}^{17} \to \mathcal{C} \cup \{\text{no facial information}\}$ maps the vector to a viewpoint category:

$$c_i = f_{\text{cls}}(\mathbf{e}_i) \tag{6}$$

$$\mathcal{C} = \{\text{front, side\_left\_front, side\_left\_back, side\_right\_front, side\_right\_back, back}\} \tag{7}$$

Frames classified as no facial information are grouped into a separate set. Hence, all video frames processed by the model can be divided into three categories:

$$\mathcal{V}_{\text{low-confidence}} = \{\mathbf{v}_i \in \mathcal{V}_{\text{final}} \mid \text{conf}(b_i) < \theta\} \tag{8}$$

$$\mathcal{V}_{\text{no facial information}} = \{\mathbf{v}_i \mid c_i = \text{no facial information}\} \tag{9}$$

$$\mathcal{V}_{\text{cat}}[c] = \{\mathbf{v}_i \mid c_i = c\}, \quad \forall c \in \mathcal{C} \tag{10}$$

Due to variations in the distribution proportions of the three types of video frames across different videos, this paper dynamically employs differentiated reference image extraction strategies tailored to each video, in order to enhance the quality of the extracted reference images. As shown in Figure 3(a), if $\mathcal{V}_{\text{cat}}[c]$ occupies a significantly higher proportion in the entire video frames than the other two categories, we prioritize selecting the frames with the most unique action from different perspective categories. The set of effective perspective categories is defined as $\mathcal{C}_{\text{exist}} \subseteq \mathcal{C}$, the number of effective perspective categories is $|\mathcal{C}_{\text{exist}}| = N_{\text{cat}}$. For each category $c \in \mathcal{C}_{\text{exist}}$, the optimal frame is:

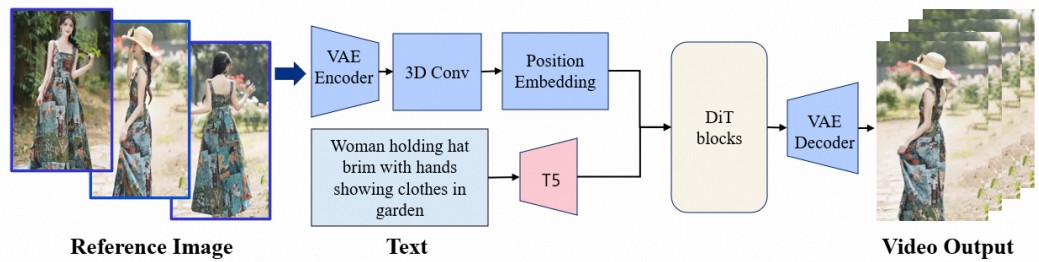

**Reference Image**     **Text**     **Video Output**

Figure 4: The framework of our method. Reference images are encoded by the VAE encoder and processed by a 3D conv to extract features. The text input is encoded by the T5 text encoder. The text and image features are then sent to the DiT blocks to generate videos.

$$\mathbf{v}_c^* = \arg \max_{\mathbf{v} \in \mathcal{V}_{\text{cat}}[c]} U(\mathbf{v}) \tag{11}$$

where $U(\cdot)$ denotes the action uniqueness scoring function. As shown in Figure 3(c), $U(\cdot)$ takes the standard pose as one of its inputs to compare the discrepancy between each pose and the standard pose. If $N_{\text{cat}} \geq 5$: Uniformly sample 5 categories $\{c_1, \ldots, c_5\}$ from $\mathcal{C}_{\text{exist}}$, and output:

$$\mathcal{V}_{\text{out}} = \left\{\mathbf{v}_{c_1}^*, \ldots, \mathbf{v}_{c_5}^*\right\} \tag{12}$$

If $N_{\text{cat}} < 5$: First include all category-optimal frames $\{\mathbf{v}_c^*\}_{c \in \mathcal{C}_{\text{exist}}}$, then supplement the remaining $5 - N_{\text{cat}}$ frames by selecting those that maximize action distance relative to the already selected set. Let $d(\mathbf{v}_i, \mathbf{v}_j)$ denote the action difference function, which is shown in Figure 3(c). The supplementary frame set is:

$$\mathcal{V}_{\text{supp}} = \arg \max_{\substack{\mathcal{S} \subseteq \bigcup_c \mathcal{V}_{\text{cat}}[c] \\ |\mathcal{S}| = 5 - N_{\text{cat}}}} \sum_{\mathbf{v} \in \mathcal{S}} \min_{\mathbf{v}^* \in \mathcal{V}^*} d(\mathbf{v}, \mathbf{v}^*) \tag{13}$$

where $\mathcal{V}^* = \{\mathbf{v}_c^*\}_{c \in \mathcal{C}_{\text{exist}}}$. Hence, the final reference images are:

$$\mathcal{V}_{\text{out}} = \left\{\mathbf{v}_c^*\right\}_{c \in \mathcal{C}_{\text{exist}}} \cup \mathcal{V}_{\text{supp}} \tag{14}$$

As shown in Figure 3 (b), if most of the frames in the video are missing facial information, we prioritizes selecting a representative frame with the most unique action from each of the front, back, and side view groups, and then supplements the remaining frames from $\mathcal{V}_{\text{no facial information}}$ through a multimodal large model (MLLM) to ensure perspective coverage and semantic. Hence, the final reference images are:

$$\mathcal{V}_{\text{out}} = \left\{\mathbf{v}_c^*\right\}_{c \in front, back, side} \cup \mathcal{V}_{\text{MLLM}} \tag{15}$$

Where $\mathcal{V}_{\text{MLLM}} = \text{MLLM}(\mathcal{V}_{\text{noface}}, N_{\text{need}})$. When the overall quality of the frequency is low, it results in low confidence in most frame detection. We delegate the frame selection task to MLLM.

## 4 APPROACH

Our objective is to construct a simple yet effective baseline framework that leverages large-scale pre-trained models to facilitate both scaling and future improvements. To this end, we build our approach upon the publicly available Wan2.2 Text-to-Video (T2V) 14B model Wan et al. (2025). As shown in Figure 4, the overall architecture integrates product-specific visual cues and positional information to ensure faithful and temporally consistent video generation. Below, we describe the two core components of our method.

### 4.1 REFERENCE IMAGE CONDITIONING

To incorporate the visual appearance of reference product images, we project each image into the latent space of the pre-trained video diffusion model without enforcing a fixed aspect ratio. Given a reference image $I_{\text{ref}} \in \mathbb{R}^{H \times W \times 3}$, the VAE encoder maps it into a latent $z_{\text{ref}} \in \mathbb{R}^{c \times h_r \times w_r}$. This latent is further processed by a lightweight 3D convolution, producing a feature map $f_{\text{ref}} \in \mathbb{R}^{c \times 1 \times h_r \times w_r}$,

where the temporal axis is explicitly included for consistency with video tokens. The feature map is flattened into a token sequence:

$$t_{\text{ref}} \in \mathbb{R}^{(h_r \times w_r) \times c}. \tag{16}$$

For multiple reference images, we repeat this process independently and concatenate all resulting sequences:

$$T_{\text{ref}} = \text{Concat}\left(t_{\text{ref}}^1, t_{\text{ref}}^2, \ldots, t_{\text{ref}}^N\right). \tag{17}$$

Since both reference and video tokens share the same embedding dimension $c$, they can be directly concatenated along the sequence dimension without additional projection. The joint token sequence is then passed into the DiT together with the text embeddings, enabling the model to integrate product-specific visual cues throughout the generation process.

## 4.2 FREQUENCY-BASED POSITIONAL EMBEDDING

To distinguish reference tokens from video tokens and to maintain spatial-temporal alignment, we adopt a frequency-based rotary positional embedding (RoPE) factorized over the frame, height, and width axes. This factorization allows the model to represent positions in a 3D coordinate system while preserving relative distances through rotational invariance. The embedding dimension is split across the three axes, and 1D RoPE bases are precomputed as:

$$\text{RoPE}_d(p) = \left[ \exp\left(i \cdot p \cdot \theta^{-2k/d}\right) \,\middle|\, k = 0, \ldots, \tfrac{d}{2} - 1 \right], \tag{18}$$

where $p$ is the position index, $d$ the axis-specific dimension, and $\theta$ a scaling constant (default $10^4$).

For a token at location $(i, j)$ in a reference feature of size $h_r \times w_r$, its positional embedding is:

$$e_{\text{pos-ref}}(i, j) = \left[\text{RoPE}_f(0), \, \text{RoPE}_h(i + h_{\text{off}}), \, \text{RoPE}_w(j + w_{\text{off}})\right], \tag{19}$$

where $h_{\text{off}}, w_{\text{off}}$ are cumulative offsets updated after each reference block ($h_{\text{off}} \leftarrow h_{\text{off}} + h_r, \; w_{\text{off}} \leftarrow w_{\text{off}} + w_r$). This ensures that tokens from different reference images and video frames occupy distinct coordinate ranges and do not overlap in the embedding space.

Finally, the positional embeddings of all reference and video tokens are concatenated:

$$E_{\text{pos}} = [E_{\text{pos-ref}}; E_{\text{pos-video}}], \tag{20}$$

and added to the corresponding token sequences before entering DiT blocks. Compared to conventional 2D sinusoidal or learned embeddings, this unified frequency-based design provides stronger structural cues across space and time, thereby improving consistency between reference images and generated video content.

## 5 EXPERIMENTS

### 5.1 METRICS

We evaluate generated videos using VBench Huang et al. (2024), a comprehensive benchmark where all metric scores are normalized to the range 0–100, with higher values indicating better performance. Below, we provide explicit definitions and interpretations for each metric employed in our evaluation.

**Appearance Metrics.** These metrics assess the visual quality and fidelity of the generated content: 1) Aesthetic Quality (0–100): Measures overall visual appeal predicted by aesthetic assessment models. 2) Image Quality (0–100): Evaluates sharpness and clarity based on perceptual quality estimation. 3) Subject Consistency (0–100): Quantifies the preservation of product appearance across frames. 4) Background Consistency (0–100): Measures stability of scene elements to detect unwanted flickering.

**Motion Metrics.** These metrics capture the temporal realism of generated motion: 1) Motion Smoothness (0–100): Assesses frame-to-frame continuity and absence of jitter. 2) Dynamic Degree (0–100): Reflects the richness and diversity of motion, penalizing overly static videos.

**Video-Condition Consistency.** These metrics evaluate how accurately the generated video adheres to the conditioning inputs: 1) Color Accuracy (0–100): Measures deviation of generated colors

Table 2: Compared results of Video quality metrics on E- CommerceVideoBench.

| Method | Appearance Metrics | | | | Motion Metrics | |
|---|---|---|---|---|---|---|
| | Aesthetic Quality ↑ | Image Quality ↑ | Subject Consistency ↑ | Background Consistency ↑ | Motion Smoothness ↑ | Dynamic Degree ↑ |
| Sora Sor (2024) | 47.2 | 49.3 | 88.1 | 94.2 | 97.2 | 88.7 |
| CogVideo5B Hong et al. (2023) | 51.4 | 53.2 | 91.6 | 95.6 | 97.5 | 87.8 |
| keling kel (2024) | 52.0 | 61.7 | 93.3 | 96.9 | 99.4 | 64.8 |
| Wan2.2 Wan et al. (2025) | 52.3 | 63.1 | 93.8 | 97.4 | 99.6 | 88.1 |
| +multi-reference | 53.4 | 64.2 | 94.8 | **98.1** | **99.7** | 89.3 |
| +position embedding | **54.7** | **65.9** | **95.6** | 98.0 | 99.6 | **90.5** |

Table 3: Compared results of Video-Condition Consistency metrics on E-CommerceVideoBench.

| Method | Object Class ↑ | Multiple Objects ↑ | Human Motion ↑ | Color ↑ | Spatial Relationship ↑ | Scene ↑ | Appearance Style ↑ | Temporal Style ↑ | Overall Consistency ↑ |
|---|---|---|---|---|---|---|---|---|---|
| Wan2.2 Wan et al. (2025) | 89.0 | 63.3 | 96.9 | 81.5 | 71.8 | 52.4 | 22.0 | 24.3 | 27.4 |
| Ours | 90.8 | 70.1 | 99.1 | 91.0 | 73.7 | 49.8 | 23.4 | 24.5 | 27.9 |

from those in the reference images. 2) Object Class (0–100): Identifies whether the product category is preserved. 3) Spatial Relationship (0–100): Evaluates the correctness of object positioning and structural relations. 4) Scene, Appearance Style, Temporal Style (all 0–100): Measure consistency of scene semantics and style attributes. 5) Overall Consistency (0–100): A composite score summarizing alignment with all conditioning signals.

These normalized and interpretable metrics allow us to comprehensively evaluate video attractiveness, motion realism, and fidelity to product attributes in e-commerce scenarios.

## 5.2 IMPLEMENTATION DETAILS

The model was trained for 5 days on 32 GPUs (4 nodes, each with 8 GPUs) using DeepSpeed. All parameters of the pre-trained DiT model were fine-tuned without freezing any modules. We adopted classifier-free guidance during training and inference to balance fidelity and diversity. We used a learning rate of $1 \times 10^{-5}$ with a linear warmup over the first 50 steps. The noise weighting scheme was set to "logsnr+uniform." The training processed videos with a maximum resolution of $1280 \times 720$ pixels, a length of 81 frames, and a target frame rate of 16 FPS.

## 5.3 MAIN RESULTS

As the quantitative results presented in the Table 2, we conduct a comprehensive analysis of video generation quality on the E-CommerceVideoBench. The compared methods are selected from the VBench leaderboard, using one reference image and text as input. The Wan2.2 baseline already demonstrates competitive performance, outperforming established methods like Sora and CogVideo5B across most appearance metrics. The incorporation of multi-reference frames brings substantial improvements, with Background Consistency reaching 98.1 and Motion Smoothness achieving 99.7. This enhancement can be attributed to the multi-reference mechanism's ability to leverage temporal information from adjacent frames, thereby improving both spatial stability and motion continuity. The position embedding strategy yields the most significant gains in Aesthetic Quality, Image Quality, and Subject Consistency. The performance improvement stems from the explicit encoding of spatial-temporal relationships, which enables the model to better maintain object permanence and structural coherence throughout the video sequence.

We also evaluate the video condition consistency metrics on VBench, with results shown in Table 3. The performance improvement stems from our multi-frame reference mechanism, enhancing temporal coherence and the position encoding, providing better spatial guidance, which collectively improves object stability and structural accuracy. We also provide some quality results in Figure 5.

| Text | Reference Image | Video |
|------|-----------------|-------|

[Indoor, Medium Shot, Eye-Level, Static Shot] An Asian woman with shoulder-length black hair stands in a room, modeling a brown outfit. She wears a long-sleeved, brown, knee-length dress with a matching cropped vest adorned with small, sparkling embellishments. The overall style is realistic and slightly commercial, emphasizing the clothing and the model's presentation.

[Concrete Wall, Medium Shot, Eye-Level, Static Shot, Tracking Shot] An Asian woman with an elegant demeanor stands against a gray concrete wall with visible bolts. She wears a maroon, knee-length, short-sleeved dress with a v-neck and a belted waist, black high-heeled sandals, black sunglasses, gold earrings, a gold watch on her left wrist, and a gold bracelet on her right wrist. She carries a small black purse with a chain strap.

[Street, Medium Shot, Eye-Level, Static Shot, Pan Right, Pan Left] The video showcases a stylish Asian man with short black hair and a goatee, wearing a black quilted jacket, a black turtleneck, black pants, and sunglasses. He also wears a watch on his left wrist and a ring on his left ring finger. He stands against a backdrop of a dark, possibly urban setting with blurred lights and a partial view of a car.

[Street, Medium Shot, Eye-Level, Tracking Shot] A young Asian woman with an elegant demeanor walks confidently down a city street. She wears a vibrant purple suit, consisting of a blazer and trousers, paired with black high-heeled shoes. A small black handbag hangs from her left hand, and a delicate gold necklace adorns her neck.

[Apartment Entrance, Medium Shot, Eye-Level, Static Shot] A young Asian woman with long, dark hair exits an apartment building through a doorway. She wears a light gray, short-sleeved crop top with crisscross straps, black high-waisted, flared pants, and black platform sandals. A black shoulder bag hangs from her left shoulder.

[White Studio, Medium Shot, Eye-Level, Static Shot, Pan Right, Pan Left] The video showcases a young woman with an Asian face, standing in a white studio against a plain white backdrop. She wears a black sports bra, a black pleated mini skirt, and white sneakers. Her dark hair is styled in two braids. Initially, she stands facing the camera, her arms relaxed at her sides.

Figure 5: Some examples generated by our method. Each example has 5 reference images input, due to the space limit, we only present 2 here.

## 6 CONCLUSION

In this paper, we address the critical need for automated e-commerce video generation by introducing E-CommerceVideo, a large-scale dataset that fills a significant gap in domain-specific video generation resources. Unlike existing general-purpose video datasets, our benchmark provides carefully curated triplets of multi-view product images, textual descriptions, and high-quality video demonstrations specifically tailored for e-commerce scenarios. Our work provides a foundation for future research in domain-specific video generation and offers practical solutions to democratize high-quality video production in e-commerce. Our future work will focus on improving the baseline method proposed in the paper.

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

# A APPENDIX

## A.1 ADAPTIVE FRAME SAMPLING (TAIL-ALIGNED STRATEGY)

---

**Algorithm 1** Adaptive Frame Sampling (Tail-Aligned Strategy)

---

**Require:** Total frames $T$, original frame rate $f_{\text{orig}}$
**Ensure:** Selected frame indices $\mathbf{F}_{\text{sel}}$

1: $T \leftarrow$ number of video frames
2: $f_{\text{sel}}^{(0)} \leftarrow \lfloor f_{\text{orig}} - 5 \rfloor$ ⊳ Initial target fps
3: $f_{\text{soft}} \leftarrow \lfloor f_{\text{orig}} \rfloor$ ⊳ Soft upper bound
4: $n_{\max} \leftarrow \lfloor 5 \cdot f_{\text{orig}} \rfloor$ ⊳ Max frame count: 5s
5: $n_{\min} \leftarrow \lfloor 3 \cdot f_{\text{orig}} \rfloor$ ⊳ Min frame count: 3s
6: $t_5 \leftarrow \lfloor 5 \cdot f_{\text{orig}} \rfloor$ ⊳ 5-second mark in frames
7: $s_{\text{start}} \leftarrow T$ ⊳ Start from tail
8: success $\leftarrow$ False
9: $n_{\text{sel}} \leftarrow -1$
10: **for** $n = n_{\max}$ **downto** $n_{\min}$ **step** 8 **do**
11:      $f'_{\text{sel}} \leftarrow \min(f_{\text{sel}}^{(0)}, f_{\text{orig}})$
12:      $\Delta t \leftarrow f_{\text{orig}} / f'_{\text{sel}}$
13:      $R \leftarrow \lfloor 1 + (n-1) \cdot \Delta t \rfloor$ ⊳ Frame cover range
14:      **if** $T - R \geq t_5$ **then**
15:          $s_{\text{start}} \leftarrow T - R$ ⊳ Tail-aligned start index
16:          $\mathbf{F}_{\text{sel}} \leftarrow \textsc{Linspace}(s_{\text{start}}, s_{\text{start}} + R - 1, n)$
17:          success $\leftarrow$ True
18:          **break**
19:      **end if**
20: **end for**
21: **if** not success **then**
22:      $n_{\text{sel}} \leftarrow n$ ⊳ Use last tried $n$
23:      **while** not success **and** $f_{\text{sel}} \leq f_{\text{soft}}$ **do**
24:          $f'_{\text{sel}} \leftarrow \min(f_{\text{sel}}, f_{\text{orig}})$
25:          $\Delta t \leftarrow f_{\text{orig}} / f'_{\text{sel}}$
26:          $R \leftarrow \lfloor 1 + (n_{\text{sel}} - 1) \cdot \Delta t \rfloor$
27:          **if** $T - s_{\text{start}} \geq R$ **then**
28:             $s_{\text{start}} \leftarrow T - R$
29:             $\mathbf{F}_{\text{sel}} \leftarrow \textsc{Linspace}(s_{\text{start}}, s_{\text{start}} + R - 1, n_{\text{sel}})$
30:             success $\leftarrow$ True
31:             **break**
32:          **end if**
33:          $f_{\text{sel}} \leftarrow f_{\text{sel}} + 1$
34:      **end while**
35: **end if**
36: **if** not success **then**
37:      skip the current video.
38: **end if**
39: **return** $\mathbf{F}_{\text{sel}}$

---

## A.2 USER STUDY

While automated metrics are valuable for quantification, they fail to capture critical subjective qualities for e-commerce, such as Logo fidelity, detailed texture preservation, and overall commercial appeal. To provide a comprehensive and impartial validation of our method's commercial viability, we conducted a large-scale human subjective evaluation.

We recruited 50 independent assessors and randomly selected 100 sets of input image/text prompts from our test set. Assessors participated in a Triple-Blind A/B/C Test, where they simultaneously

viewed videos generated by our method (Ours) and wan2.2 for the same input. The presentation order was randomized.

Assessors rated each video using a 5-point Likert Scale (1=Poor, 5=Excellent) based on three key commercial criteria:

1) Product Fidelity: Measures the accuracy of color, Logo details, and texture consistency relative to the input image. 2) Motion Quality: Measures the smoothness, naturalness, and effectiveness of product movement. 3) Commercial Appeal: Measures the video's overall professionalism and attractiveness as an e-commerce advertisement.

The results in Table 4 show that our proposed method significantly outperformed the baselines across all subjective metrics, achieving a clear advantage in the commercially critical aspects of Product Fidelity and overall Appeal. This strongly validates our method's effectiveness in generating high-quality, commercially ready e-commerce videos.

Table 4: User Study Results (5-point Likert Scale)

| Method | Fidelity (F-Score) | Motion Quality (M-Score) | Appeal (A-Score) |
|---|---|---|---|
| Wan2.2 | 3.68 | 3.85 | 3.59 |
| **Ours** | **4.31** | **4.10** | **4.25** |

