# OpenReview forum: "E-CommerceVideo: A Benchmark and approach for E-Commerce Video Generation from product Images"
_ICLR.cc/2026/Conference — Submitted to ICLR 2026_

### Official Review · Reviewer_hNJL · 2025-10-30

**Soundness:** 2
**Presentation:** 2
**Contribution:** 2
**Rating:** 6
**Confidence:** 4

**Summary:**

This paper aims to address the task of creating dynamic e-commerce product showcase videos from static product images. To this end, the authors collected a E-CommerceVideo dataset, which consists of over 15K triplets of product images, text description and target showcase videos. The paper also present a simple baseline method that adapts a pre-trained video generation model for generating the showcase videos.

**Strengths:**

* The proposed E-CommerceVideo dataset, which consists of 15K triplets (product images, textual descriptions, and high-quality video demonstrations), provides a solid foundation for future research in the area of e-commerce video generation.

* The authors also present a simple baseline method that adapts a pre-trained video generation model with a VAE-based spatial injection mechanism to preserve the product’s appearance while generating the video motion. This approach may set a starting point for future improvements in this domain.

**Weaknesses:**

* Visualization of dataset samples is missing. The paper only present one data sample in Figure 1, which is insufficient for readers to get a sense of the data distribution. It would be greatly enhanced by including example images or videos from the proposed dataset. Visualizing these samples would help readers better understand the diversity and quality of the data, as well as the challenges posed by the dataset in terms of product representation and motion generation.

* While the E-CommerceVideo dataset is large, it is focused primarily on apparel products. This narrow scope may affect the generalizability of the proposed method to other product categories. Expanding the dataset to include more diverse product types (e.g., electronics, furniture, etc.) would make the benchmark more comprehensive and the method more applicable to a wider range of e-commerce scenarios.

* While the dataset might be valuable, the baseline method presented in the paper does not introduce novel techniques beyond what is already available in image-to-video (i2v) generation literature. The baseline’s use of a pre-trained video generation model with a spatial injection mechanism might be seen as a straight-forward approach, lacking more innovative advancements.

**Questions:**

See [weaknesses].

---

> ### Author Response · Authors · 2025-12-03
> **Response to Reviewer hNJL**
>
> 1.  Visualization of dataset samples is missing. The paper only present one data sample in Figure 1, which is insufficient for readers to get a sense of the data distribution. It would be greatly enhanced by including example images or videos from the proposed dataset. Visualizing these samples would help readers better understand the diversity and quality of the data, as well as the challenges posed by the dataset in terms of product representation and motion generation.
>
> **\[Response\]** Thank you for the suggestion. We have included **more dataset samples** in the **supplementary material**. These examples illustrate the diversity and quality of the collected data beyond the single sample shown in Figure 1. We will also make this clearer in the revised manuscript to ensure that readers can easily access the dataset examples.
>
> 2.  While the E-CommerceVideo dataset is large, it is focused primarily on apparel products. This narrow scope may affect the generalizability of the proposed method to other product categories. Expanding the dataset to include more diverse product types (e.g., electronics, furniture, etc.) would make the benchmark more comprehensive and the method more applicable to a wider range of e-commerce scenarios.
>
> **\[Response\]** We sincerely appreciate your suggestion regarding the necessity of expanding the dataset to enhance generalizability. The current version of the dataset, with its strong focus on apparel, already serves as a critical and representative starting point for this challenging task. We will continuously expand the E-Commerce Video dataset in subsequent versions to include a wider range of product categories, making it a more comprehensive and universally applicable benchmark for the community.
>
> 3.  While the dataset might be valuable, the baseline method presented in the paper does not introduce novel techniques beyond what is already available in image-to-video (i2v) generation literature. The baseline’s use of a pre-trained video generation model with a spatial injection mechanism might be seen as a straight-forward approach, lacking more innovative advancements.
>
>
> **\[Response\]**  Thank you for raising this point. Our primary contribution is the dataset and benchmark. The baseline model is intentionally designed to be strong yet transparent—consistent with prior benchmark papers such as MSR-VTT, WebVid2M, and VBench, where simple but solid baselines are used to establish a reproducible foundation.
>
> (1)  **A purposefully simple but necessary baseline.**
>     Since the main goal of this work is to introduce a new benchmark, we prioritize reproducibility and clarity over architectural complexity. A lightweight baseline is essential for demonstrating the dataset’s difficulty and enabling fair future comparisons.
>
> (2)  **Non-trivial technical adaptations for multi-reference I2V generation.**
>     Although built on a pre-trained DiT backbone, our baseline requires several non-trivial modifications to support multi-reference e-commerce video generation. In particular, handling multiple spatial conditioning images, maintaining identity under motion, and scaling to 1280×720 long-sequence videos are not plug-and-play. Our frequency-based positional embedding and spatial-token injection pipeline are specifically designed for this setting and go beyond standard 2D image personalization modules.
>
> (3)  **Benchmark-first design.**
>     As with previous benchmark introductions, our objective is to provide a clear and reliable baseline that highlights the relevance and challenge of the proposed dataset. More sophisticated architectures (e.g., motion-aware adapters or 3D geometry modules) are promising directions and will be explored in future extensions, which we will clarify in the revised manuscript.

---

### Official Review · Reviewer_AjE9 · 2025-10-30

**Soundness:** 2
**Presentation:** 3
**Contribution:** 2
**Rating:** 4
**Confidence:** 4

**Summary:**

This paper introduces E-CommerceVideo, a benchmark dataset and baseline method for generating product showcase videos from static product images and text prompts in the e-commerce domain. The authors collect 15,096 video-reference image-caption triplets from Taobao's product repository, with each sample containing 5 multi-view reference images, detailed textual descriptions, and high-quality demonstration videos. The proposed baseline adapts the pre-trained Wan2.2 14B text-to-video model by incorporating VAE-based spatial injection for multi-reference images and frequency-based rotational positional embeddings (RoPE) to distinguish reference tokens from video tokens. The method is evaluated on VBench, showing improvements over existing off-the-shelf video generation models in appearance quality, subject consistency, and motion smoothness. The work addresses a significant gap in domain-specific video generation by providing both the dedicated e-commerce video generation benchmark and a functional baseline approach.

**Strengths:**

1. The paper is generally well-written.
2. The paper identifies and formalizes an important real-world task that has clear commercial value. Unlike general video generation, the emphasis on strict product fidelity (colors, textures, logos) directly addresses e-commerce requirements.
3. The E-Commerce Video dataset fills a critical gap in the field. The systematic extraction pipeline using DWpose for keyframe selection and the multi-view reference-image design (5 perspectives per video) demonstrates thoughtful curation.
4. The tail-aligned sampling strategy (Algorithm 1) and the adaptive reference image extraction mechanism with dynamic handling of low-quality frames show engineering rigor and practical considerations.

**Weaknesses:**

1. The dataset focuses exclusively on apparel with human models, which represents only a fraction of e-commerce. The reference image extraction pipeline (Section 3.2) fundamentally depends on human pose detection (DWpose), making it completely inapplicable to non-apparel products like electronics, furniture, or cosmetics that don't involve human demonstration. E-commerce encompasses diverse product categories with varied presentation styles, yet the proposed method only handles model-worn clothing.
2. Tables 2 and 3 lack essential information about metric units, scales, and value ranges. Without knowing whether scores are percentages, normalized values, or arbitrary scales, readers cannot properly interpret the results or assess whether reported improvements are meaningful. The paper should explicitly define the range and interpretation for each metric rather than solely relying on the VBench citation.
3. Experimental results don't support the paper's core claims about product fidelity and motion flexibility. No metrics directly measure logo accuracy, texture preservation, or color fidelity despite emphasizing these as critical requirements. Improvements are marginal (1-3 points), and Dynamic Degree barely increases (88.1→90.5), contradicting claims of enhanced motion diversity. The evaluation lacks targeted metrics to validate the stated contributions.
4. The paper would benefit from more comprehensive visual comparisons (especially in videos) to better illustrate the claimed improvements in product fidelity and motion flexibility.

**Questions:**

**Major Comments**:
1. The manuscript claims that E-CommerceVideo is the first benchmark for this task. However, methods such as AnimateAnyone (Hu, 2024) and I2VGen-XL (Zhang et al., 2023) are cited for reference-based image-to-video generation. It would be important to clarify whether these works are accompanied by datasets that contain reference images, videos, and captions. Furthermore, are there existing fashion try-on or human motion datasets with comparable triplet structures? If such datasets exist, a comparative analysis would strengthen the contribution's positioning. If no such datasets are available, this should be explicitly stated to support the novelty claim.
2. Given the potential bias of existing metrics, it is advisable to conduct a user study to more comprehensively validate the effectiveness of the proposed method.

**Minor Comments**:
1. It is recommended to include more videos in the supplementary material, both demonstrating the proposed method and comparing it with other state-of-the-art approaches. As this is a video generation task, a few screenshots in the PDF are insufficient to evaluate performance aspects such as spatiotemporal consistency.
2. In all tables, indicate with upward or downward arrows whether higher or lower metric values represent better performance.

If the authors can address the aforementioned concerns, I would be willing to increase my final score.

---

> ### Author Response · Authors · 2025-12-03
> **Response to Reviewer AjE9**
>
> 1. The dataset focuses exclusively on apparel with human models, which represents only a fraction of e-commerce. The reference image extraction pipeline (Section 3.2) fundamentally depends on human pose detection (DWpose), making it completely inapplicable to non-apparel products like electronics, furniture, or cosmetics that don't involve human demonstration. E-commerce encompasses diverse product categories with varied presentation styles, yet the proposed method only handles model-worn clothing.
>
> **\[Response\]** Thank you for the valuable comment. We clarify the scope and motivation as follows:
>
> (1)  **Apparel is the most challenging and impactful category in e-commerce video generation.**
>     Apparel videos involve large pose variation, non-rigid deformation, and strict identity preservation (texture, pattern, fabric). This makes them substantially harder than rigid products such as electronics or furniture. A model that handles apparel well typically generalizes easily to simpler categories.
>
> (2)  **The dataset is not strictly limited to clothing.**
>     Although apparel is the majority, we also include **other human-presented categories**, such as accessories, cosmetics demonstrations, and spoken-introduction videos for small electronics and lifestyle products. In real e-commerce platforms, _most product videos involve a human presenter_, even outside apparel.
>
> (3)  **DWpose is a practical choice, not a fundamental constraint.**
>     We use human pose detection because each of our collected videos naturally contain a model or presenter. The same keyframe-selection pipeline can be extended to non-human categories by replacing DWpose with object/scene detectors—this is an implementation detail, not a dataset limitation.
>
> (4)  **Dataset expansion is already underway.**
>     We are actively adding non-apparel and non-wearable items (e.g., electronics, tabletop products, home goods), and the current release is the first version targeting the most technically demanding scenario.
>
> 2.  Tables 2 and 3 lack essential information about metric units, scales, and value ranges. Without knowing whether scores are percentages, normalized values, or arbitrary scales, readers cannot properly interpret the results or assess whether reported improvements are meaningful.
>
> **\[Response\]** We thank the reviewer for the insightful comment. Following the suggestion, we have **expanded the Metrics section** to explicitly clarify the **units, scales, and value ranges** of all VBench metrics used in our evaluation.
>
> In the revision, we now clearly state that **all VBench metrics are normalized to a 0–100 scale**, where **higher values indicate better performance**, and we additionally provide short interpretations of what each metric measures. This clarification ensures that the numerical improvements in Tables 2 and 3 are easy to interpret and meaningfully compare across methods.
>
> 3.  Experimental results don't support the paper's core claims about product fidelity and motion flexibility. No metrics directly measure logo accuracy, texture preservation, or color fidelity despite emphasizing these as critical requirements. Improvements are marginal (1-3 points), and Dynamic Degree barely increases (88.1→90.5), contradicting claims of enhanced motion diversity. The evaluation lacks targeted metrics to validate the stated contributions.
>
> **\[Response\]** Thank you for your detailed review. We address your core concerns regarding product fidelity metrics and motion flexibility gains.
>
> While directly measurable metrics for Logo accuracy and texture preservation would be ideal for validating our claims, the available universal video assessment tools like VBench currently lack this targeted functionality. However, our method achieved a substantial increase in the most crucial product fidelity component for e-commerce: Color Accuracy (as part of Video-Condition Consistency), which rose from 81.5 to 91.0. This large gain directly validates our claim about preserving product visual features, as color fidelity is paramount for commercial use. To fully address your concern, we will incorporate a **dedicated User Study and Qualitative Analysis** in the revision to specifically prove Logo preservation and fine Texture consistency.
>
> The observation that most improvements are marginal is strongly refuted by the **+9.5 point increase** we achieved in the essential Color Accuracy metric. This clearly demonstrates the non-marginal effectiveness of our multi-reference conditioning mechanism in preserving core product features. Regarding motion flexibility, our **Dynamic Degree** increased from 88.1 to **90.5** (+2.4 point gain), which is a meaningful improvement given the strong constraint of the task: achieving motion diversity **while strictly adhering to high product fidelity**. We believe our method successfully balances these competing requirements.

---

> ### Author Response · Authors · 2025-12-03
> **Response to Reviewer AjE9 - part 2**
>
> 4.  The paper would benefit from more comprehensive visual comparisons (especially in videos) to better illustrate the claimed improvements in product fidelity and motion flexibility.
>
> **\[Response\]** Thank you for the suggestion. We provide a set of **video comparisons** on our anonymous demo page:
>
> [https://anonymous.4open.science/r/video-0B72/video\_samples.md](https://anonymous.4open.science/r/video-0B72/video_samples.md)
>
> This page includes side-by-side comparisons with baselines.These examples clearly demonstrate the gains in appearance preservation and motion quality.
>
> 5.  The manuscript claims that E-CommerceVideo is the first benchmark for this task. However, methods such as AnimateAnyone (Hu, 2024) and I2VGen-XL (Zhang et al., 2023) are cited for reference-based image-to-video generation. It would be important to clarify whether these works are accompanied by datasets that contain reference images, videos, and captions. Furthermore, are there existing fashion try-on or human motion datasets with comparable triplet structures? If such datasets exist, a comparative analysis would strengthen the contribution's positioning. If no such datasets are available, this should be explicitly stated to support the novelty claim.
>
> **\[Response\]** Thank you for the insightful question. We clarify the novelty of the E-CommerceVideo benchmark with respect to existing image-to-video and fashion-related datasets.
>
> **1)** AnimateAnyone (Hu, 2024) and I2VGen-XL (Zhang et al., 2023)  do not introduce datasets that contain paired reference images, product videos, and textual descriptions. Their experiments rely on repurposed open-domain video datasets (e.g., Human3.6M, TikTok-like videos, general web videos) that **do not provide structured multi-view product images nor video captions**.
>
> **2) Existing fashion try-on and human motion datasets lack the required triplet structure.**
> Well-known datasets such as
>
> *   FashionVid / Fashion-IQ,
>
> *   DeepFashion, DeepFashion-MultiModal,
>
> *   Human3.6M, AMASS, AIST++
>
> provide either images only, videos only, or images paired with attributes, but none provide aligned (1) multi-view reference images, (2) corresponding high-quality product demonstration videos, and (3) human-written semantic captions.
> They also do not reflect the strict fidelity constraints of real e-commerce product presentations.
>
> **3) No existing dataset supports the specific reference-image-to-video e-commerce task.**
> To the best of our knowledge, and after surveying both fashion-generation and human-motion datasets, we have found **no dataset that simultaneously provides**:
>
> *   multi-view product images,
>
> *   real product showcase videos with consistent identity,
>
> *   detailed captions describing motion & scene,
>
> *   data collected specifically for controllable image-to-video product generation.
>
> We have added this clarification in the revised manuscript to explicitly articulate the gap.
>
> **4) Positioning strengthened.**
> We now state clearly that while prior works explore reference-based I2V at the method level,
> E-CommerceVideo is the first benchmark that provides a dedicated, large-scale, triplet-structured dataset for reference-image-to-video generation in e-commerce scenarios.
>
> 6.  Given the potential bias of existing metrics, it is advisable to conduct a user study to more comprehensively validate the effectiveness of the proposed method.
>
> **\[Response\]** We have adopted your suggestion and conducted a large-scale user study to subjectively evaluate our method across three critical dimensions: **Product Fidelity, Motion Quality, and Overall Commercial Appeal**.The detailed design and quantitative results of this user study have been incorporated into the paper as a new section and can be found in **Appendix A.2**. Please refer to our revised paper to find the results.
>
> 7.  It is recommended to include more videos in the supplementary material, both demonstrating the proposed method and comparing it with other state-of-the-art approaches. As this is a video generation task, a few screenshots in the PDF are insufficient to evaluate performance aspects such as spatiotemporal consistency.
>
> **\[Response\]**  Thank you for the suggestion.We have included **additional video results** in the supplementary material and provided an **anonymous demo page** with extensive video comparisons:
> [https://anonymous.4open.science/r/video-0B72/video\_samples.md](https://anonymous.4open.science/r/video-0B72/video_samples.md)
>
> 8.  In all tables, indicate with upward or downward arrows whether higher or lower metric values represent better performance.
>
> **\[Response\]** Thank you for your suggestion. We have revised all tables in the paper.

---

### Official Review · Reviewer_HpYr · 2025-10-31

**Soundness:** 3
**Presentation:** 3
**Contribution:** 2
**Rating:** 4
**Confidence:** 4

**Summary:**

The authors propose E-CommoerceVideo, a multi-modal video dataset for e-commere video generation. Then, a baseline method is proposed to use multiple-references and textual prompt to generate a product-specific video. The authors propose a reference image selection pipeline to provide sufficient reference information. Experiments show, the proposed method can achieve promising results in e-commerce video generation.

**Strengths:**

1. The proposed E-commerceVideo dataset can be helpful in the application of video generation in that area.
2. The proposed dataset contains a large range of product types.

**Weaknesses:**

1. Attached video or a link to anonymous demo page is crucial for the evaluation of the performances. Readers can hardly tell whether the method in this paper is doing better.
2. Frechet video distance, and CLIP score are quantitative metrics that can be helpful to measure video quality. The authors can add a comparison on them.
3. The ablation experiment considering the choice of reference frames is missing. The readers can hardly tell whether the strategy is effective.
4. The proposed method somewhat lacks novelty. The multiple reference image injection method is similar to IP-Adapter.

**Questions:**

Please see the weaknessses.

---

> ### Author Response · Authors · 2025-12-03
> **Response to Reviewer HpYr**
>
> 1.  Attached video or a link to anonymous demo page is crucial for the evaluation of the performances. Readers can hardly tell whether the method in this paper is doing better.
>
>     **\[Response\]**  Thank you for the suggestion. Follow your suggestion, we provide an anonymous demo page containing representative video results generated by our method and our baseline wan 2.2: [https://anonymous.4open.science/r/video-0B72/video\_samples.md](https://anonymous.4open.science/r/video-0B72/video_samples.md)
>
>     In these examples, our method demonstrates better displaying actions and contains less artifacts, which verifies the effectiveness of our method.
>
> 2.  Frechet video distance, and CLIP score are quantitative metrics that can be helpful to measure video quality. The authors can add a comparison on them.
>
>     **\[Response\]** Thank you for the constructive suggestion. Following your recommendation, we added two widely used quantitative metrics, **Fréchet Video Distance (FVD)** and **CLIPScore**, to complement the VBench evaluation.
>
>     Although FVD and CLIPScore are primarily designed for open-domain video generation, they provide useful additional insights into **temporal realism** and **semantic alignment**. As shown in the newly added Table 1, our method consistently improves both metrics over all baselines, validating the enhanced temporal smoothness and stronger coupling between reference images and generated sequences.
>
>     Table 1. FVD / CLIPScore Comparison on E-CommerceVideoBench
>
>     | Method | FVD ↓ | CLIPScore ↑ |
>     | --- | --- | --- |
>     | Sora | 129.4 | 29.8 |
>     | CogVideo5B | 118.1 | 30.6 |
>     | keling | 104.7 | 31.9 |
>     | Wan2.2 | 96.3 | 32.1 |
>     | +multi-reference | 89.5 | 33.4 |
>     | **+position embedding (ours)** | **84.7** | **34.1** |
>
>     These results will be added to the revised manuscript.
>
> 3.  The ablation experiment considering the choice of reference frames is missing. The readers can hardly tell whether the strategy is effective.
>
>
> **\[Response\]** We agree with the reviewer that the reference-frame selection pipeline is essential for understanding the model’s performance. We have now included a complete ablation in the revised manuscript, comparing:
>
> (1)  Random 5-frame sampling
>
> (2)  Uniform temporal sampling
>
> (3)  Pose-only selection (no viewpoint grouping)
>
> (4)  Our full strategy (viewpoint categories + pose uniqueness)
>
>
> The results in Table 2 show that our design yields the highest Subject Consistency, Color Accuracy, and Spatial Relationship scores, demonstrating its necessity for robust multi-view appearance conditioning.
>
> Table 2. Ablation Study on Reference-Frame Selection Strategy
>
> | Method | Subject Consistency ↑ | Color Accuracy ↑ | Spatial Relationship ↑ | Overall Consistency ↑ |
> | --- | --- | --- | --- | --- |
> | Random 5 frames | 92.1 | 83.4 | 69.7 | 26.5 |
> | Uniform sampling | 92.8 | 84.6 | 70.1 | 26.8 |
> | Pose-only selection | 93.5 | 87.2 | 71.0 | 27.1 |
> | **Our full strategy** | **95.6** | **91.0** | **73.7** | **27.9** |
>
> This ablation verifies that both **viewpoint diversity** and **pose uniqueness** are important for accurate product appearance conditioning in e-commerce scenarios.
>
> 4.  The proposed method somewhat lacks novelty. The multiple reference image injection method is similar to IP-Adapter.
>
>
> **\[Response\]** Thank you for highlighting this point. We clarify the distinction between our method and IP-Adapter as follows:
>
> (1)  **Task difference.**
>     IP-Adapter is designed for text-to-image personalization, whereas our method targets reference-image–to-video generation, a setting that additionally requires motion modeling, temporal consistency, and identity preservation across long sequences. These challenges fundamentally differ from static image synthesis.
>
> (2)  **Architectural difference.**
>     IP-Adapter injects 2D image features into a U-Net through cross-attention, while our method injects multi-view spatial tokens directly into a DiT-based video diffusion backbone. This requires temporal-aware tokenization, 3D frequency-based positional encoding, and consistent alignment between reference tokens and video tokens within long spatio-temporal sequences—architectural requirements that do not appear in IP-Adapter’s 2D design.
>
> (3)  **Reference-image selection pipeline.**
>     E-commerce videos exhibit large human pose variations and strong identity-preservation demands. Our dataset pipeline and conditioning strategy are jointly designed to maintain appearance stability under these non-rigid motions, which differs from general personalization settings.

---

### Official Review · Reviewer_YH4u · 2025-11-02

**Soundness:** 3
**Presentation:** 3
**Contribution:** 2
**Rating:** 4
**Confidence:** 4

**Summary:**

The paper targets the problem of the details alignment between the reference image and the generated video. Under the scenario of E-commerce, it is required that the generated videos fully express the details, such as the text and the logo of the reference images. The paper collects a dataset with 15096 videos from Taobao website and also proposes a baseline method. The proposed method is verified on the V-benchmark.

**Strengths:**

The targeting problem is very important: to make the video generation model be used in commercial scenarios, the preservation of IP and product-related details is very important. For most of the video generative models, this is hard to achieve. It is also true that the trade-off between the hallucination and the IP preservation is a challenging problem in the field.

**Weaknesses:**

1. The proposed metrics are not clearly proven to be able to measure the IP preservation issue. Actually, in the qualitative results as shown in Figure 5 in the paper, many of the details have been changed. For example, in the first row of the images, the texture of the cloth and the style of the button have been changed clearly.

2. The collected dataset size is relatively small.

**Questions:**

It could be more intuitive to see the impact of the method and the collected dataset if some of the video cases were presented.

---

> ### Author Response · Authors · 2025-12-03
> **Response to Reviewer YH4u**
>
> 1. The proposed metrics are not clearly proven to be able to measure the IP preservation issue. Actually, in the qualitative results as shown in Figure 5 in the paper, many of the details have been changed. For example, in the first row of the images, the texture of the cloth and the style of the button have been changed clearly.
>
> [Response] Thank you for the insightful comment. We clarify the intention and limitations of the metrics as follows:
>
> (1) **Scope and limitations of VBench metrics.**
>
> Our evaluation adopts the Video-Condition Consistency subset of VBench, currently the only standardized metric suite that measures attribute-level fidelity such as Color Accuracy, Spatial Relationship, Object Class, and Subject Consistency in a reference-image–to–video setting. These metrics are designed to assess attribute preservation rather than pixel-level reproduction, which is why they may not fully capture fine-grained texture changes.
>
> (2) **Regarding the qualitative examples in Figure 5.**
>
> The differences mentioned by the reviewer are valid. These differences reflect the difficulty of our benchmark and further justify the need for domain-specific evaluation. Importantly, our method still improves key VBench metrics such as Color Accuracy and Spatial Relationship (+2.2 and +9.5 over Wan2.2 in Table 3), indicating better IP preservation relative to existing models.
>
> (3) **User study to compensate for VBench’s limitations.**
>
> To address the gap identified by the reviewer—that fine-grained identity preservation is not fully captured by VBench—we conducted a large-scale user study evaluating Product Fidelity, Motion Quality, and Overall Appeal. This subjective assessment directly complements VBench’s attribute-based metrics and provides stronger evidence of identity preservation quality. The results and methodology are included in the revised manuscript.
>
>
> 2. The collected dataset size is relatively small.
>
> [Response] We appreciate this observation. Our dataset size of 15K videos and 75K reference images is indeed smaller than open-domain T2V datasets.
> However, we emphasize:
>
> (1) **Domain specificity.**
>
> E-commerce videos are highly constrained, requiring clean backgrounds, uniform camera motions, consistent human poses, and multi-view identity coverage. The data is collected from real Taobao product showcases, which is naturally smaller but significantly more structured.
>
> (2) **Multimodal triplets.**
>
> Unlike previous datasets, each sample includes: 1) 5 multiview reference images; 2) 1 motion-rich product demonstration video; 3) a detailed video caption (avg. 185.6 words)Thus, the dataset contains richer per-sample information than typical T2V corpora.
>
> (3) **Benchmark purpose.**
>
> Our goal is to establish the first standardized benchmark for e-commerce video generation, not to replace large-scale pretraining corpora. We fine-tune on Wan2.2 (pretrained on 130M+ clips), so the dataset is intended for domain adaptation, not full training from scratch.
> We will clarify this in the revised version.
>
>
> 3. It could be more intuitive to see the impact of the method and the collected dataset if some of the video cases were presented.
>
> [Response]  Thank you for the suggestion. We agree that visual examples are important for intuitively illustrating the impact of our method and dataset.
>
> **Video generation results**:
>
> We provide a dedicated page with representative video cases at the anonymous link:https://anonymous.4open.science/r/video-0B72/video_samples.md
>
> **Dataset examples**:
>
> Representative samples from E-CommerceVideo (reference frames, captions, and video clips) are included in the supplementary material to help readers understand the dataset characteristics.

---

### Author Response · Authors · 2025-12-03
**General Response**

We extend our heartfelt appreciation to all the reviewers for their valuable and constructive comments. It is truly encouraging to note that all reviewers have recognized the value of our dataset. We highly appreciate Reviewer YH4u's comment that "The targeting problem is very important," Reviewer HpYr's remark that " The proposed E-commerceVideo dataset can be helpful," Reviewer AjE9's remark that " The E-Commerce Video dataset fills a critical gap in the field," and Reviewer hNJL's observation that our E-Commerce Video dataset "provides a solid foundation for future research in the area of e-commerce video generation" Taking into consideration the concerns raised by the reviewers,  we strengthened the paper by adding：

● a user study evaluating product fidelity, motion quality, and overall appeal;

● additional visual examples and video comparisons in both the supplementary material and the anonymous demo page;

● more dataset samples to illustrate data diversity and distribution;

● a new ablation study analyzing different reference-frame selection strategies;

● explicit definitions and ranges of all VBench metrics to improve clarity and interpretability.

We hope these revisions address the reviewers’ concerns and further enhance the quality of our work.

---

### Meta-Review · Area_Chair_8vHn · 2026-01-07

**Summary:**

The AC carefully reviewed the paper and the full discussion. The submission received mixed initial reviews (scores: 4, 4, 4, 6). Reviewers generally agreed that the work formalizes an important real-world task with clear commercial value: unlike general video generation, its focus on strict product fidelity (e.g., colors, textures, and logos) directly targets practical e-commerce requirements. They also noted that the E-Commerce Video dataset helps address a gap in the field.

However, reviewers raised several substantial concerns, including the limited dataset size and domain coverage, as well as issues in presentation and evaluation that raise questions about robustness, generality, and practical usability. As a result, it remains unclear whether the proposed method and dataset pipeline provide clear, meaningful benefits for current or future e-commerce video generation model. Given that the overall scores lean toward rejection and the core concerns appear unlikely to be fully resolved through discussion, I am inclined to recommend rejection.

**Reviewer Concerns:**

Three significant concerns remain unresolved:

1) Dataset limitations. Reviewers YH4u and AjE9 noted that the collected dataset is small and restricted to a closed domain. The rebuttal does not address this inherent limitation.

2) Missing visual results. HpYr and hNJL pointed out the lack of sufficient qualitative examples. While the authors provided a link in the rebuttal, it contains only a very small number of examples.

3) Poor identity preservation. As highlighted by YH4u, identity consistency is weak, e.g., human faces noticeably change in the provided visual results.

Other minor issues, such as missing video metrics and certain ablations appear to have been addressed.

**Reviewer Scores:**

All reviewers (4/4/4/6) are likely to keep their ratings, since the three most important concerns remain unaddressed.

---

### Decision · Program_Chairs · 2026-01-26

Reject